# Shelf Life Prediction and Kinetics of Quality Changes in Pineapple (*Ananas comosus*) Varieties at Different Storage Temperatures

**Maimunah Mohd Ali** [1] **, Norhashila Hashim** [1,2,*] **, Samsuzana Abd Aziz** [1,2] **and Ola Lasekan** [3]

1    Department of Biological and Agricultural Engineering, Faculty of Engineering, Universiti Putra Malaysia, Serdang 43400, Selangor, Malaysia
2    SMART Farming Technology Research Centre, Faculty of Engineering, Universiti Putra Malaysia, Serdang 43400, Selangor, Malaysia
3    Department of Food Technology, Faculty of Food Science and Technology, Universiti Putra Malaysia, Serdang 43400, Selangor, Malaysia
*    Correspondence: norhashila@upm.edu.my; Tel.: +603-97694336; Fax: +603-89466425

**Abstract:** Shelf life estimation is an important factor to predict the freshness of fruits. This study aimed to investigate the shelf life and the changes in the physicochemical properties of three different pineapple varieties, namely MD2, Josapine, and Morris stored at 5, 10, and 25 °C. The effect of storage temperature on pineapple changes in total soluble solids, pH, moisture content, firmness, and colour was evaluated for 21 days of the storage period. It was revealed that different storage temperatures have a significant effect on the shelf life and quality of different pineapple varieties. The firmness and moisture content showed high regression coefficients, hence were used for the shelf life prediction of pineapple based on kinetic models. By using first-order kinetics, the coefficient of determination ($R^2$) values for quality changes in pineapples ranged from 0.893 to 0.992. The results also demonstrated that the samples stored at 10 °C had the longest shelf life in relation to the changes in firmness and moisture content of the fruit. The findings indicated that shelf life estimation plays an important role to improve the quality preservation of fresh fruits and vegetables during storage.

**Keywords:** pineapple; fruit quality; shelf life; storage; kinetic model

## 1. Introduction

Pineapple (*Ananas comosus*) is one of the most popular tropical fruits worldwide with exceptional flavour and nutritional value. In 2019, the top five pineapple production countries in the world were Costa Rica (3328.10 tonnes), the Philippines (2747.86 tonnes), Brazil (2426.53 tonnes), Indonesia (2196.46 tonnes), and China (1727.61 tonnes) [1]. The fruit is a good source of dietary fibre, antioxidants, vitamins, and minerals, including vitamin A and ascorbic acid [2,3]. The fruit is also a non-climacteric fruit that is consumed for its pleasant taste and possesses many essential health-promoting benefits. There are many types of pineapple varieties with varied shapes, sizes, colours, and flavours, resulting in a hybrid variety in an effort to increase the market demand. Pineapples of different varieties differ in terms of nutritional and physicochemical properties.

Generally, the quality of pineapple is assessed in terms of the physicochemical properties and shelf life of the fruit during postharvest storage. These include the texture, flavour, appearance, and chemical composition of the fruit that could influence consumer acceptability and preference [4]. Several aspects such as postharvest handling and storage temperature could affect the quality and shelf life, as well as the sensory characteristics of the fruit during storage [5,6]. A low storage temperature has been proven to extend the shelf life and freshness of fresh produce by reducing the ripening process after being harvested [7]. Although the ripening process is reduced by a low temperature, other factors

such as humidity, temperature, and water activity could also lead to fruit quality deterioration during storage [8]. During this period, the fruit can be ripened quickly, resulting in quality loss and a shortened shelf life. The quality and shelf life determination of pineapple heavily relies on conventional methods that are quite laborious and time-consuming [9,10].

Numerous studies have investigated the quality evaluation of pineapple under various postharvest conditions including sorting, grading, drying, thermal processing, freezing, canning, etc. Hartono et al. [11] examined the influence of temperature on the physical properties of pineapple in terms of skin colour, hardness, and weight loss of the fruit. George et al. [12] determined the physical and physiological attributes of pineapple at five different growth stages after anthesis and recorded a major reduction in the firmness, pH, and ascorbic acid content. Furthermore, Ismail et al. [13] reported the use of microwave-assisted processing for assessing the total soluble solids (TSS), pH, as well as water content of pineapple fruit. Leneveu-jenvrin et al. [14] investigated the effects of freezing treatment on the microbiological and physicochemical properties of Queen Victoria pineapple, which resulted in significant variations in quality defects and colour changes.

The shelf life of fruits and vegetables is of great importance as it has remarkable value in terms of the nutritional, chemical, and sensory properties during postharvest handling [15]. Normally, the shelf life is determined according to the deterioration level of the quality attributes of fruit developed during storage. The development of predictive models to estimate fruit shelf life is required to monitor the changes in physicochemical properties and ensure its marketability. In this sense, several kinetic models have been established to create an acceptability limit for the prediction of quality attributes and shelf life of food [16,17]. Monitoring the storage temperature is a critical factor in predicting the quality changes in fruit during storage. However, existing research focused mostly on the processed products of pineapples but not on fresh pineapple fruits. Wanakamol and Poonlarp [18] determined the predicted shelf life of vacuum-fried pineapple chips based on the changes in colour and rancidity when stored at 30 °C. Montero-Calderón et al. [19] reported the shelf life prediction of pineapple slices using different packaging conditions. Gómez et al. [20] evaluated the changes in firmness and colour of pineapple slices in equilibrium-modified atmosphere packaging in order to predict the shelf life of the fruit. Chakraborty et al. [21] studied the predicted shelf life of pineapple puree at different storage conditions. There is a lack of information which focused on the kinetics and shelf life prediction for fresh pineapple fruits during storage. Hence, it is crucial to explore the knowledge in the aspect of postharvest storage in order to avoid fruit losses. The present work aims to develop a shelf life prediction model for pineapple stored at three different storage temperatures by analysing the quality attributes of the fruit during storage using kinetic models. Considering the effect of different storage conditions, it is important to determine the optimal shelf life and kinetic parameters of pineapple fruit that could serve as baseline data in future.

## 2. Materials and Methods

### 2.1. Sample Preparation

Pineapple fruit varieties MD2, Josapine, and Morris were obtained from an orchard in Simpang Renggam, Johor, Malaysia. All the pineapple fruits were harvested on the same day to avoid the seasonal variances in the physicochemical properties between the varieties from the same cultivation area and transported immediately to the Biomaterials Processing Laboratory, Universiti Putra Malaysia, after harvest. Fruit samples at a maturity level index of 2 were harvested to evaluate the quality changes in the fruit for a period of three weeks. Fruit samples at a maturity level index of 2 were harvested to evaluate the shelf life for a period of three weeks. At this maturity level, the fruit samples were 50% unripe, glossy dark green, and had traces of yellow colour between eyes at the base. These pineapple varieties were chosen based on homogeneity in size, shape, and grade according to the Federal Agricultural Marketing Authority (FAMA). The selected fruit had a medium size and 1st-grade quality with an average weight of 1.0–1.2 kg. All pineapple samples

were cleaned from dirt, weighed, and labelled before storage. The samples were stored at selected storage temperatures based on these three storage conditions: chilling temperature (5 °C), cold storage condition (10 °C), and ambient temperature (25 °C) after being stored at room temperature for 24 h at 85 to 90% relative humidity. The fruit samples were randomly grouped at four storage time intervals (Day 0, Day 7, Day 14, and Day 21) which consisted of 30 samples for each storage day. In this study, 120 samples were randomly selected at each storage temperature for each pineapple variety accumulating to a total of 1080 fresh fruit samples. For each physicochemical property, 30 samples from each storage day were measured with three replications. The three different varieties of pineapple before storage (Day 0) and at the end of storage (Day 21) subjected to different storage temperatures are shown in Figure 1.

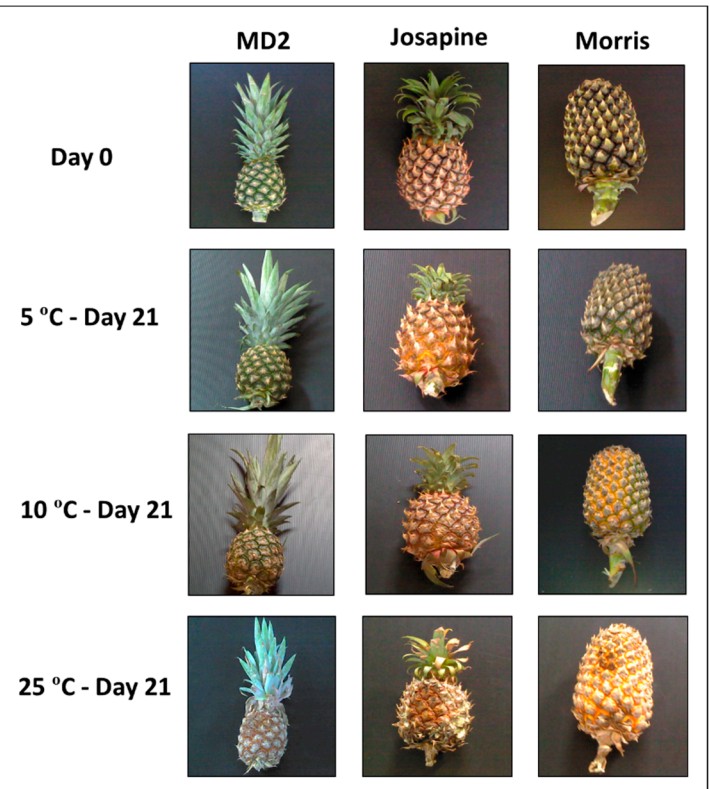

**Figure 1.** The images of different varieties of pineapple before storage (Day 0) and at the end of storage (Day 21) subjected to different storage temperatures.

### 2.2. Determination of Physicochemical Properties

#### 2.2.1. Total Soluble Solids

The TSS content was measured from the pineapple juice using a digital refractometer (Pal-1, Atago Co., Tokyo, Japan) after filtering through Whatman paper No. 1. The TSS content was expressed as a percentage and calculated from the average of three measurements.

#### 2.2.2. pH

10 g of pineapple flesh was homogenised with a buffer solution (pH 7) to extract the juice. The pH of the pineapple juice was measured using a pH meter (DPH-2, Atago Co., Tokyo, Japan). Three measurements were obtained for each pineapple sample as units of pH.

#### 2.2.3. Firmness

The firmness of pineapple flesh was determined using a penetrometer (GY-1, G-tech Co., Ltd., Guangdong, China) with a plunger tip measuring 3.5 mm in diameter. The

pineapple firmness was assessed at three different parts of the pineapples (top, middle, and bottom). The maximum force was exerted on the pineapple flesh to obtain the average value from the samples.

### 2.2.4. Moisture Content

The moisture content was determined using an oven-drying method at 105 °C until a constant weight was reached. The pineapple cube (3 cm³) was placed in a metal dish and dried in an air-drying oven (WS-301, Tsung Hsing Food Machinery, Kaohsiung City, Taiwan). The moisture content was calculated as a percentage based on weight.

### 2.2.5. Colour Evaluation

The surface colour of pineapple flesh was evaluated using a colourimeter (NR20XE, Shenzhen 3nh Technology, Shenzhen, China) with a 20 mm measuring aperture. Colour values were determined using the CIE L* a* b* scale and calibrated with a white reference tile before conducting the measurements. The colour values were defined as L* (brightness), a* (redness/greenness), and b* (yellowness/blueness), respectively. The colour measurements were taken at three different locations (top, middle, and bottom) to calculate the average value for pineapple flesh.

### 2.3. Kinetic Model Development

For food storage, the Arrhenius law is the most widely used model in food-related tasks, abided by the zero and first-order kinetic models because temperature correlates to the reaction rate [22]. The general rate law for describing the quality changes in the fruit is shown in Equation (1). The zero and first-order kinetic models are defined in Equations (2) and (3):

$$\frac{dCt}{dt} = -kC_t^n \tag{1}$$

$$Zero\ order = Ct = Co - kt \tag{2}$$

$$First\ order = Ct = Co\ \exp(-kt) \tag{3}$$

where $C_t$ is the initial value of physicochemical properties, $k$ is the reaction rate, $n$ is the reaction order, and $t$ is the storage time.

The temperature dependence of the reaction of the physicochemical properties was defined by the Arrhenius equation [10]. The Arrhenius model is shown in Equation (4).

$$\ln k = \ln kref - \frac{Ea}{R}\left(\frac{1}{T} - \frac{1}{Tref}\right) \tag{4}$$

where $k_{ref}$ is the reaction rate at reference temperature (day$^{-1}$), $E_a$ is the activation energy (kJ/mol), $R$ is the gas constant (8.314 J/mol K), $T$ is the absolute temperature (K), and $T_{ref}$ (K) is the reference temperature.

### 2.4. Statistical Analysis

Data for the physicochemical properties of pineapple samples were analysed using analysis of variance (ANOVA). Multiple mean comparisons between physicochemical properties on each storage day with different temperatures were performed by using Tukey's test at a 5% significance level. Both ANOVA and multiple mean comparisons were calculated using SAS software (Version 9.4, SAS Institute, Cary, NC, USA). The performance of the fitted models was determined by the coefficient of determination ($R^2$) and root mean square error (RMSE).

### 3. Results and Discussion

*3.1. Effect of Storage Temperature on TSS, pH, Firmness, and Moisture Content*

The means comparison of all the physicochemical properties of pineapples, including TSS, pH, firmness, and moisture content, on the storage days for the different pineapple varieties was analysed using Tukey's test. The values that are represented by different letters within the same column show significant differences between the storage days at different storage temperatures for all physicochemical properties of the fruit. Considering the fact that pineapple is a non-climacteric fruit, the quality changes in the fruit vary and are not uniform. Generally, different pineapple varieties have different unique traits and characteristics. For this reason, pineapples are evaluated based on the physicochemical attributes of different varieties with acceptable flavour and morphological characteristics. Table 1 shows the changes in TSS values of three different pineapple varieties during 21 days of storage. Based on the results, the TSS values of pineapple varieties (MD2, Josapine, and Morris) ranged from 3.60 to 16.60%. The range of TSS values was significantly higher than those reported by Dolhaji et al. [7], which ranged from 8.8 to 16.0%, due to the conversion of starch to sugar during the softening process of the fruit. From this current study, Josapine had the highest TSS values on Day 14 (25 °C), which obtained a range of 9.40 to 16.60%. In contrast, MD2 recorded TSS values ranging from 9.40 to 15.60%, followed by Morris (3.60–10.10%), respectively.

**Table 1.** Changes in total soluble solids and pH of pineapples at different storage temperatures during storage.

| Temperature | Day | MD2 | | Josapine | | Morris | |
|---|---|---|---|---|---|---|---|
| | | TSS (%) | pH | TSS (%) | pH | TSS (%) | pH |
| 5 °C | 0 | 10.71 ± 0.04 [a] | 3.00 ± 0.11 [b] | 10.10 ± 0.84 [b] | 2.80 ± 1.02 [a] | 3.60 ± 0.74 [ab] | 3.20 ± 1.39 [a] |
| | 7 | 13.50 ± 1.48 [ab] | 2.90 ± 0.01 [a] | 12.30 ± 1.16 [ab] | 2.90 ± 0.92 [b] | 8.20 ± 0.54 [a] | 3.00 ± 0.07 [b] |
| | 14 | 15.34 ± 0.13 [b] | 2.40 ± 0.07 [ab] | 14.10 ± 0.03 [a] | 2.60 ± 0.97 [c] | 9.40 ± 0.02 [b] | 2.70 ± 1.29 [c] |
| | 21 | 14.60 ± 0.64 [c] | 2.60 ± 0.48 [c] | 12.73 ± 0.16 [c] | 2.40 ± 0.96 [ab] | 8.00 ± 0.16 [c] | 2.50 ± 1.04 [ab] |
| 10 °C | 0 | 11.30 ± 0.54 [a] | 3.00 ± 0.39 [ab] | 12.20 ± 0.25 [a] | 3.00 ± 1.04 [b] | 9.20 ± 0.04 [a] | 2.90 ± 1.03 [b] |
| | 7 | 12.10 ± 0.26 [b] | 2.80 ± 0.16 [b] | 12.50 ± 0.95 [ab] | 2.90 ± 0.03 [a] | 9.70 ± 0.42 [b] | 2.70 ± 0.36 [a] |
| | 14 | 13.70 ± 0.92 [ab] | 2.70 ± 0.49 [a] | 13.10 ± 0.22 [b] | 2.90 ± 0.38 [a] | 10.01 ± 1.82 [ab] | 2.50 ± 0.64 [ab] |
| | 21 | 11.40 ± 0.02 [c] | 2.60 ± 1.467 [c] | 11.80 ± 0.19 [c] | 2.70 ± 0.26 [ab] | 7.60 ± 0.05 [c] | 2.60 ± 0.02 [c] |
| 25 °C | 0 | 10.10 ± 0.16 [a] | 3.30 ± 0.16 [a] | 9.40 ± 0.02 [a] | 4.10 ± 0.73 [b] | 7.80 ± 0.16 [a] | 3.60 ± 1.36 [b] |
| | 7 | 11.20 ± 0.07 [ab] | 3.20 ± 0.85 [b] | 14.70 ± 0.74 [ab] | 4.00 ± 0.05 [ab] | 9.80 ± 0.21 [ab] | 2.90 ± 0.83 [a] |
| | 14 | 15.60 ± 0.26 [b] | 3.10 ± 0.01 [c] | 16.60 ± 0.26 [c] | 3.10 ± 0.36 [a] | 10.10 ± 0.94 [ac] | 2.70 ± 1.06 [ab] |
| | 21 | 9.40 ± 0.85 [a] | 2.90 ± 0.74 [ab] | 10.70 ± 0.02 [b] | 2.80 ± 0.02 [c] | 8.40 ± 0.21 [b] | 2.70 ± 0.17 [c] |

Results are presented as mean ± standard deviation. The values that are represented by different letters within the same column show significant differences between the storage days at different storage temperatures using Tukey's test at $p < 0.05$.

It was demonstrated that the TSS values show overlapping between Day 0 and Day 7, whereas a significant difference was found between Day 14 and Day 21 for all pineapple varieties. The trend of TSS values increased gradually from Day 0 until Day 14 and showed a decrease at the end of storage (Day 21) for MD2, Josapine, and Morris, respectively. It was observed that a storage temperature of 25 °C had the highest TSS value compared with the pineapples stored at 5 and 10 °C, except for the Morris variety. The increase in TSS during postharvest storage was related to the storage time as well as the respiration process in the fruits [23,24]. The highest TSS values among all storage temperatures were achieved at 25 °C for Josapine (16.60%) and MD2 (15.60%), whereas Morris recorded TSS values of 10.10% at 25 °C. In addition, it can be seen that the TSS values of pineapples were affected by storage temperature and time. Nevertheless, the storage time at which the changes took place varied in terms of storage temperature depending on the chemical and biochemical properties present in pineapple during storage. Monitoring the storage

temperature is essential to ensure the optimum conditions for minimising fruit damage and extending the shelf life of the fruit.

The pH values of three different pineapple varieties (MD2, Josapine, and Morris) during storage ranged from 2.40 to 4.10 (Table 1). These results were significantly lower than those reported by Siti Rashima et al. [25] who obtained pH values of pineapples ranging from 3.70 to 4.30, which might have been affected by microbial activities during storage. In addition, Padrón-Mederos et al. [4] found that the pH values decreased considerably during 10 days of storage. Based on the results, all pineapple varieties had the highest pH values recorded on Day 0 (25 °C), which was 4.10 for Josapine, followed by Morris (3.60) and MD2 (3.30), respectively. The pH values of pineapples declined with fruit maturity, in the acidic range of 3.3 to 3.8, which was in line with the presented findings in [26]. Moreover, it can be observed that storage at a temperature of 5 °C had the lowest pH values compared with the pineapples stored at 10 and 25 °C for the varieties MD2 and Josapine, whereas for the variety Morris, the lowest pH values were obtained at 10 °C. For the storage temperature of 5 °C, the fruit was susceptible to chilling injury which could lead to deterioration of the fruit quality. A gradual decrease in pH values during storage from Day 0 to Day 21 was observed for all pineapple varieties at different storage temperatures. As a result, lower pH values were found at the end of the storage day, which indicated the spoilage of the fruit during the storage period. A similar trend was obtained for all storage temperatures, which is mainly associated with the respiratory metabolic activity of the fruits [27].

The firmness values of three different pineapple varieties (MD2, Josapine, and Morris) during storage ranged from 0.33 to 2.92 N (Table 2). Padrón-Mederos et al. [4] observed a gradual decline in the textural properties of pineapples in cold storage which could be linked to pectin degradation. These results were in accordance with Siow and Lee [28], who reported the firmness values of pineapples ranging from 0.20 to 3.50 N. Based on the results, Morris had the highest firmness values on Day 0 (10 °C), which was 2.92 N. Among all the pineapple varieties, Morris recorded huge drops in terms of firmness values, especially at 25 °C compared with 5 and 10 °C. In a similar manner, a gradual decrease in firmness values was detected from Day 0 to Day 21, which could imply that the textural characteristics of the fruit decreased as the storage days increased. Similarly, the decrease in the firmness of pineapples was reported along the maturity process in such a way that fruit growth contributed to the accumulation of water content [12]. This consequence occurred due to the fact that the texture development of the pineapples changed accordingly with the maturation pattern of the fruit after harvest [29]. The changes in textural properties indicated the corresponding variation in the internal quality attributes of the fruit along with the ripening process. In the current study, the higher firmness values could be an indicator of fruit immaturity at harvest. Mature pineapple will ripen properly with a pleasant flavour and taste, but fruit harvested at the immature stages does not ripen properly and results in poor quality. The composition of pineapple flesh might also vary between different varieties of the fruit. In addition, the harvest maturity and time, fruit variety, and environmental conditions also influence the quality [30]. Thus, the evaluation of quality attributes of different pineapple varieties is vital to ensure the fruit is within an acceptable quality range.

The moisture content values of three different pineapple varieties (MD2, Josapine, and Morris) during three weeks of storage ranged from 68.87 to 95.26% (Table 2). The results obtained were comparable with the data reported by Padrón-Mederos et al. [4] using the Red Spanish variety of pineapple, with moisture content ranging from 84 to 89%. Based on the findings, Josapine had the highest moisture content values on Day 21 (25 °C), which was 95.26%. On the other hand, Morris recorded content values ranging from 68.87 to 92.85%, followed by MD2 (84.93–91.27%), respectively. The results also demonstrated the variation in terms of moisture content for all pineapple varieties, with the exception of Morris stored at 5 °C in which a slight drop was obtained on Day 21 (90.39%). The high moisture content in pineapples in cold storage strongly implies that the fruit might be prone to microbial

spoilage arising from the growth of microorganisms [31]. Thus, it may be inferred that the moisture content of the pineapple varieties was hugely influenced by the storage treatment, which could be an important factor to extend the shelf life and storage of the fruit. It can be noted that despite the fact that no sensory analysis was conducted, the fruits were still edible for up to 21 days when stored at the optimum storage temperature.

**Table 2.** Changes in firmness and moisture content of pineapples during storage at different storage temperatures.

| Temperature | Day | MD2 | | Josapine | | Morris | |
|---|---|---|---|---|---|---|---|
| | | Firmness (N) | Moisture Content (%) | Firmness (N) | Moisture Content (%) | Firmness (N) | Moisture Content (%) |
| 5 °C | 0 | 1.48 ± 0.89 [a] | 84.93 ± 0.22 [a] | 1.39 ± 0.16 [a] | 86.88 ± 1.26 [a] | 1.59 ± 0.73 [a] | 90.35 ± 1.26 [b] |
| | 7 | 1.43 ± 0.58 [b] | 85.72 ± 0.84 [b] | 1.01 ± 0.06 [b] | 90.76 ± 0.68 [ab] | 1.47 ± 0.93 [b] | 91.75 ± 0.01 [a] |
| | 14 | 1.23 ± 0.06 [c] | 87.82 ± 1.83 [bc] | 0.90 ± 0.03 [c] | 91.58 ± 1.04 [b] | 1.40 ± 0.03 [ab] | 92.85 ± 1.15 [c] |
| | 21 | 1.15 ± 0.93 [bc] | 89.52 ± 0.03 [c] | 0.83 ± 0.14 [ab] | 93.66 ± 1.84 [b] | 1.31 ± 0.18 [c] | 90.39 ± 1.35 [ab] |
| 10 °C | 0 | 1.48 ± 1.63 [a] | 86.93 ± 0.26 [a] | 1.45 ± 0.02 [a] | 85.56 ± 0.36 [b] | 2.92 ± 0.03 [a] | 68.87 ± 2.25 [b] |
| | 7 | 1.01 ± 0.05 [ab] | 88.78 ± 0.89 [b] | 0.63 ± 0.52 [ab] | 87.27 ± 0.39 [a] | 2.47 ± 0.15 [a] | 70.72 ± 1.94 [b] |
| | 14 | 0.80 ± 0.16 [b] | 89.50 ± 0.12 [c] | 0.61 ± 0.25 [b] | 91.39 ± 1.82 [c] | 2.49 ± 0.74 [a] | 81.48 ± 1.16 [a] |
| | 21 | 0.62 ± 0.05 [ac] | 91.08 ± 0.84 [bc] | 0.55 ± 0.07 [bc] | 92.86 ± 0.28 [bc] | 1.17 ± 0.15 [b] | 89.95 ± 1.54 [b] |
| 25 °C | 0 | 1.47 ± 1.87 [a] | 85.25 ± 0.36 [a] | 1.63 ± 0.26 [a] | 85.67 ± 0.15 [b] | 1.27 ± 0.14 [a] | 84.07 ± 1.06 [b] |
| | 7 | 1.51 ± 0.75 [b] | 87.52 ± 0.07 [ab] | 1.44 ± 0.14 [ab] | 89.15 ± 0.64 [a] | 0.77 ± 0.06 [ab] | 86.15 ± 0.38 [a] |
| | 14 | 0.68 ± 0.65 [ab] | 90.47 ± 0.03 [b] | 1.05 ± 1.66 [c] | 92.83 ± 0.20 [ab] | 0.65 ± 1.53 [b] | 87.69 ± 0.02 [ab] |
| | 21 | 0.33 ± 1.03 [c] | 91.27 ± 0.15 [a] | 0.92 ± 0.13 [b] | 95.26 ± 0.03 [c] | 0.45 ± 0.04 [ab] | 89.66 ± 0.17 [c] |

Results are presented as mean ± standard deviation. The values represented by different letters within the same column show significant differences between storage days at different storage temperatures using Tukey's test at $p < 0.05$.

### 3.2. Effect of Storage Temperature on Colour Parameters

Colour changes in pineapple varieties at different storage temperatures are shown in Figure 2. In terms of colour changes, the a* and b* values increased significantly, whereas the L* values decreased during storage for all storage temperatures. The flesh colour of pineapple is one of the main indicators for evaluating the freshness of the fruit based on the preference of the consumer. The chemical reaction in perishable fruit occurred during postharvest storage, which leads to variation in colour changes caused by chlorophyll breakdown and fruit pigments [32]. Based on the findings, it was demonstrated that Josapine and MD2 obtained the highest and similar L* values of 58.39 at 25 °C, followed by Morris (43.28) at 5 °C, respectively. A significant reduction ($p < 0.05$) in the L* values over the storage days was observed for all pineapple varieties at different storage temperatures. The differences in the lightness of the flesh colour noticeably denoted that different pineapple varieties possessed distinct characteristics and quality attributes. These findings were in agreement with those for pineapples stored for one week, with L* values that varied from 33.68 to 56.46 and an upward trend for each maturity index [26]. Temperature and storage time were vital in controlling the colour degradation of pineapples during the storage period with respect to the textural properties of the fruit [33,34].

Notably, the increase in a* values was also affected by the fruit softening during storage. Based on the findings, it was revealed that Josapine obtained a* values of 5.27 to 14.37, followed by MD2 (0.08–12.82) and Morris (0.92–11.83), respectively. A significant increase ($p < 0.05$) in the a* values over the storage days was observed for all pineapple varieties at different storage temperatures. The storage temperature at 25 °C had the lowest a* values compared with the pineapple samples stored at 5 and 10 °C, except for the Josapine variety. The highest a* value among all storage temperatures was achieved at 10 °C for Josapine (14.37), whereas MD2 and Morris recorded 12.82 and 11.83 at 25 °C, respectively. In this manner, the a* values increased significantly with storage day and temperature. The rise in a* values during storage that was observed could be related to fruit translucency and

internal browning under different temperatures [35]. The a* colour parameter of minimally processed pineapples showed a similar trend, indicating the brownish colour scheme was noticeable in fruit flesh after 8 days of storage [27].

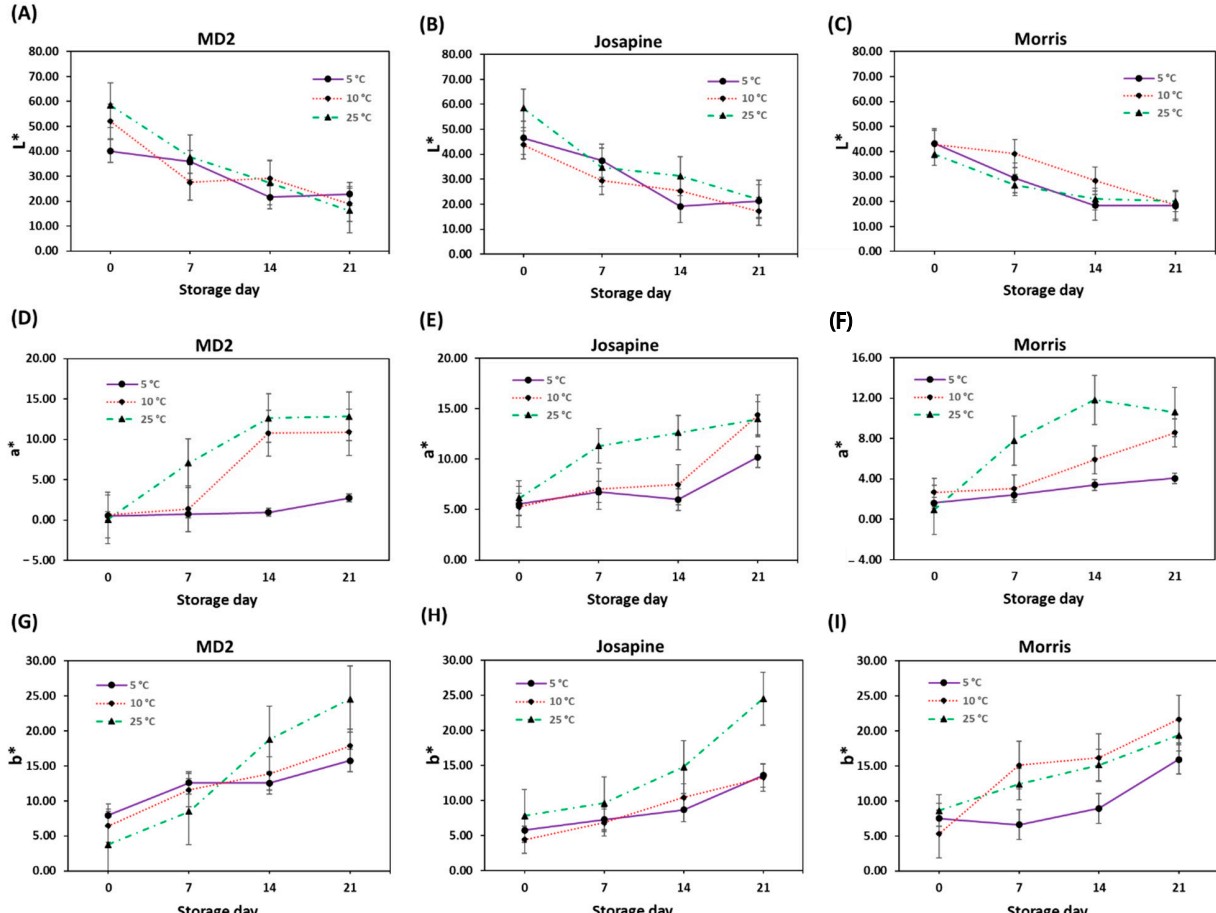

**Figure 2.** Colour changes in (**A**) L* for MD2, (**B**) L* for Josapine, (**C**) L* for Morris, (**D**) a* for MD2, (**E**) a* for Josapine, (**F**) a* for Morris, (**G**) b* for MD2, (**H**) b* for Josapine, and (**I**) b* for Morris at different storage temperatures.

Considering the effect of storage temperature on the b* values, an increasing pattern ($p < 0.05$) was demonstrated across the storage days for all pineapple varieties. Based on the findings, it was revealed that Josapine obtained the b* values of 4.40 to 25.54, followed by MD2 (3.76–24.54) and Morris (5.27–21.65). The storage temperature of 10 °C had the lowest b* values compared with the pineapple samples stored at 5 and 25 °C, except for the MD2 variety. After 14 days of storage, the b* values gradually increased, indicating that the pineapples changed to a darker colour and induced luminosity loss. The b* parameter discolouration was generated by internal bruising and physiological disorders, which include enzymatic browning in the fruits [36]. Identically, the increase in b* colour values for pineapples with the taste and texture of the fruit flesh was evaluated, indicating that unripe fruit possessed a sour flavour and firm texture, whereas overripe fruit denoted a soft texture and tangy flavour [25]. Therefore, the variation in terms of L*, a*, and b* colour parameters of different pineapple varieties in relation to different storage treatments could provide robust information which is beneficial for monitoring the functional and quality attributes of the fruit.

### 3.3. Kinetics of Quality Changes in Pineapples

The kinetics of physicochemical changes in pineapples, including TSS, pH, moisture content, firmness, and colour (L*, a*, and b*), were evaluated in order to calculate the

change rate of these parameters. The reaction order estimation achieved from fitting the zero and first-order kinetic models for TSS, pH, firmness, and moisture content of different pineapple varieties is shown in Table 3. The reaction order of quality changes in pineapple varieties was determined according to the $R^2$ and RMSE as a function of storage days at three different storage temperatures. Based on the findings, the physicochemical properties fitted better with the first-order kinetic models, with $R^2$ values from 0.893 to 0.992 and RMSE values of 0.032–3.959, respectively. Meanwhile, the zero-order kinetic models obtained $R^2$ values from 0.872 to 0.988 and RMSE values from 0.038 to 3.648. It can be noted that the $R^2$ values were slightly lower for the zero-order kinetic models compared with the first-order kinetic models. The findings signified that the changes in the physicochemical properties of pineapples demonstrated relatively good performance and fitted to first-order kinetics based on the evaluation of $R^2$ and RMSE values. Niu et al. [16] developed kinetic models to evaluate the shelf life prediction of mushrooms (*Flammulina velutipes*) based on sensory evaluations and microbial infection at three different temperatures (4, 15, and 25 °C).

**Table 3.** Reaction order estimation for physicochemical properties of different pineapple varieties during storage.

| Variety | Quality Indices | Temperature (°C) | Zero-Order | | | First-Order | | |
|---|---|---|---|---|---|---|---|---|
| | | | k | $R^2$ | RMSE | k | $R^2$ | RMSE |
| MD2 | Total soluble solids | 5 | −0.025 | 0.915 | 1.962 | −0.006 | 0.925 | 0.986 |
| | | 10 | −0.014 | 0.893 | 0.506 | −0.003 | 0.910 | 1.582 |
| | | 25 | 0.085 | 0.894 | 0.963 | 0.007 | 0.902 | 3.959 |
| | pH | 5 | 0.026 | 0.909 | 1.156 | 0.018 | 0.911 | 1.863 |
| | | 10 | −0.011 | 0.911 | 0.849 | −0.016 | 0.928 | 1.479 |
| | | 25 | −0.056 | 0.882 | 0.252 | −0.099 | 0.899 | 0.960 |
| | Firmness | 5 | 0.068 | 0.936 | 0.056 | 0.005 | 0.948 | 2.066 |
| | | 10 | 0.157 | 0.932 | 0.084 | 0.022 | 0.946 | 0.042 |
| | | 25 | 0.186 | 0.926 | 0.969 | −0.001 | 0.973 | 0.068 |
| | Moisture content | 5 | 0.266 | 0.942 | 0.263 | −0.003 | 0.964 | 1.964 |
| | | 10 | 0.005 | 0.946 | 1.642 | 0.002 | 0.978 | 0.485 |
| | | 25 | 0.047 | 0.969 | 1.859 | 0.095 | 0.985 | 0.958 |
| Josapine | Total soluble solids | 5 | −0.003 | 0.884 | 0.857 | −0.002 | 0.899 | 1.435 |
| | | 10 | −0.016 | 0.911 | 0.854 | −0.007 | 0.923 | 0.645 |
| | | 25 | 0.025 | 0.872 | 0.234 | 0.007 | 0.893 | 0.658 |
| | pH | 5 | −0.096 | 0.901 | 3.648 | −0.002 | 0.921 | 2.543 |
| | | 10 | −0.019 | 0.925 | 1.074 | −0.001 | 0.955 | 0.125 |
| | | 25 | −0.026 | 0.919 | 1.532 | −0.001 | 0.909 | 0.643 |
| | Firmness | 5 | 0.218 | 0.936 | 1.525 | 0.002 | 0.943 | 1.353 |
| | | 10 | 0.005 | 0.973 | 0.094 | 0.005 | 0.982 | 0.032 |
| | | 25 | 0.233 | 0.926 | 0.252 | −0.005 | 0.951 | 0.943 |
| | Moisture content | 5 | 0.095 | 0.973 | 1.524 | −0.001 | 0.992 | 3.545 |
| | | 10 | 0.250 | 0.988 | 0.043 | −0.002 | 0.991 | 0.245 |
| | | 25 | 0.464 | 0.930 | 0.524 | 0.001 | 0.956 | 1.352 |
| Morris | Total soluble solids | 5 | −0.053 | 0.908 | 0.069 | −0.003 | 0.921 | 0.385 |
| | | 10 | −0.059 | 0.915 | 0.958 | −0.009 | 0.920 | 0.589 |
| | | 25 | 0.068 | 0.904 | 1.849 | 0.029 | 0.938 | 0.596 |
| | pH | 5 | 0.003 | 0.877 | 0.038 | 0.001 | 0.911 | 0.591 |
| | | 10 | 0.258 | 0.889 | 2.842 | 0.048 | 0.948 | 1.106 |
| | | 25 | −0.524 | 0.900 | 0.153 | −0.027 | 0.947 | 0.859 |
| | Firmness | 5 | 0.025 | 0.893 | 2.597 | −0.006 | 0.924 | 0.058 |
| | | 10 | 0.170 | 0.921 | 0.296 | −0.009 | 0.963 | 0.472 |
| | | 25 | 0.219 | 0.928 | 1.110 | 0.024 | 0.947 | 0.859 |
| | Moisture content | 5 | 0.058 | 0.919 | 0.307 | −0.008 | 0.934 | 0.285 |
| | | 10 | 0.188 | 0.899 | 0.253 | −0.086 | 0.902 | 0.297 |
| | | 25 | 0.048 | 0.934 | 0.069 | 0.003 | 0.989 | 0.059 |

k = reaction rate; $R^2$ = coefficient of determination; RMSE = root mean square error.

The reaction order estimation achieved from fitting the zero- and first-order kinetic models for the colour parameters of pineapple varieties is shown in Table 4. Most of the colour parameters (L*, a*, and b*) had a tendency to demonstrate a high degradation of colour with storage day and temperature, implying high activation energy during storage [37]. The first-order reaction model is commonly used in many existing studies and successfully applied to monitor quality changes in various food products [38,39]. In addition, the values for rate constants fluctuated for all the physicochemical properties at 5, 10, and 25 °C, which specified that the quality changes occurred rapidly in pineapples in the chilling injury condition in comparison with the storage temperatures. The fluctuation in the rate constant values indicated the model precisely accounted for the changes in physicochemical properties over storage temperatures.

**Table 4.** Reaction order estimation for colour parameters of different pineapple varieties during storage.

| Variety | Colour Parameter | Temperature (°C) | Zero-Order | | | First-Order | | |
|---|---|---|---|---|---|---|---|---|
| | | | k | $R^2$ | RMSE | k | $R^2$ | RMSE |
| MD2 | L* | 5 | 0.064 | 0.924 | 0.030 | 0.004 | 0.901 | 0.495 |
| | | 10 | 0.269 | 0.895 | 0.096 | 0.002 | 0.921 | 1.489 |
| | | 25 | 0.053 | 0.896 | 1.856 | 0.024 | 0.899 | 0.948 |
| | a* | 5 | −0.266 | 0.911 | 1.382 | −0.007 | 0.879 | 1.859 |
| | | 10 | −0.832 | 0.890 | 0.789 | −0.263 | 0.932 | 0.083 |
| | | 25 | 0.063 | 0.932 | 0.356 | 0.006 | 0.924 | 1.489 |
| | b* | 5 | 0.025 | 0.899 | 0.598 | 0.004 | 0.921 | 0.968 |
| | | 10 | 0.028 | 0.925 | 1.056 | 0.008 | 0.927 | 0.257 |
| | | 25 | 0.053 | 0.924 | 0.859 | 0.002 | 0.935 | 0.585 |
| Josapine | L* | 5 | 0.021 | 0.875 | 0.343 | 0.001 | 0.892 | 2.437 |
| | | 10 | 0.763 | 0.949 | 0.032 | 0.002 | 0.934 | 1.242 |
| | | 25 | −0.004 | 0.868 | 0.521 | −0.004 | 0.890 | 0.654 |
| | a* | 5 | −0.847 | 0.872 | 2.520 | −0.012 | 0.888 | 1.435 |
| | | 10 | −0.598 | 0.911 | 0.345 | −0.046 | 0.925 | 0.743 |
| | | 25 | 0.558 | 0.884 | 0.824 | 0.017 | 0.886 | 0.422 |
| | b* | 5 | 0.957 | 0.884 | 1.252 | 0.002 | 0.889 | 0.867 |
| | | 10 | 0.484 | 0.879 | 0.245 | 0.001 | 0.897 | 0.079 |
| | | 25 | 0.638 | 0.875 | 0.522 | 0.004 | 0.890 | 0.243 |
| Morris | L* | 5 | 0.042 | 0.885 | 0.085 | 0.002 | 0.899 | 0.396 |
| | | 10 | 0.068 | 0.895 | 0.964 | 0.003 | 0.906 | 0.396 |
| | | 25 | 0.146 | 0.904 | 1.396 | 0.042 | 0.921 | 2.853 |
| | a* | 5 | −0.047 | 0.914 | 0.842 | −0.024 | 0.922 | 0.252 |
| | | 10 | −0.385 | 0.911 | 0.078 | −0.059 | 0.914 | 0.963 |
| | | 25 | 0.004 | 0.895 | 2.954 | 0.023 | 0.896 | 2.953 |
| | b* | 5 | 0.003 | 0.885 | 0.021 | 0.009 | 0.915 | 0.085 |
| | | 10 | 0.001 | 0.907 | 0.528 | 0.003 | 0.932 | 0.389 |
| | | 25 | 0.002 | 0.918 | 0.953 | 0.005 | 0.920 | 0.958 |

k = reaction rate; $R^2$ = coefficient of determination; RMSE = root mean square error.

The kinetic parameters of physicochemical properties of different pineapple varieties during storage obtained by the Arrhenius model are shown in Table 5. The activation energies ($E_a$) ranged from 34.793 to 36.896 kJ/mol and $k_{ref}$ ranged from 0.094 to 1.929 day$^{-1}$. A kinetic study was described using the rate constant and $R^2$ values to evaluate the quality changes and shelf life of frozen spinach, including vitamin C, chlorophyll contents, colour properties, texture, as well as sensory characteristics [40]. Apart from that, the applicability of the thermal inactivation of mangosteen was tested using Arrhenius kinetic and Weibull models in order to predict polyphenol oxidase inactivation and colour evaluation of fruit at temperatures from 60 to 100 °C [41]. The prediction of quality parameters of tomatoes was investigated in order to prolong the shelf life up to 30 days using a temperature of 10 °C [42].

Likewise, supporting findings have been reported which showed that the kinetic constants increased with the rise in temperature from 5 °C to 35 °C for the shelf life estimation of yoghurt during a storage period of 25 days [10]. The changes in fruit quality during storage, including chemical, physical, and physicochemical properties, could have an effect on the kinetics.

**Table 5.** Kinetic parameters of physicochemical properties of different pineapple varieties during storage obtained by the Arrhenius equation.

| Variety | Physicochemical Properties | $E_a$ (kJ/mol) | $k_{ref}$ (day$^{-1}$) | $R^2$ | RMSE |
|---|---|---|---|---|---|
| MD2 | TSS | 35.856 | 0.425 | 0.892 | 1.524 |
| | pH | 35.647 | 0.873 | 0.921 | 0.043 |
| | Firmness | 34.934 | 1.464 | 0.909 | 0.645 |
| | Moisture content | 36.352 | 1.929 | 0.946 | 0.658 |
| | L* | 35.067 | 0.472 | 0.931 | 0.046 |
| | a* | 35.679 | 0.643 | 0.948 | 0.422 |
| | b* | 34.974 | 0.174 | 0.927 | 0.854 |
| Josapine | TSS | 35.656 | 0.246 | 0.935 | 0.234 |
| | pH | 35.175 | 0.247 | 0.933 | 0.079 |
| | Firmness | 35.675 | 1.094 | 0.992 | 0.243 |
| | Moisture content | 36.896 | 1.549 | 0.989 | 0.743 |
| | L* | 35.680 | 0.892 | 0.951 | 0.422 |
| | a* | 35.935 | 0.218 | 0.907 | 0.043 |
| | b* | 34.793 | 0.094 | 0.935 | 0.524 |
| Morris | TSS | 35.969 | 0.137 | 0.918 | 0.343 |
| | pH | 35.159 | 0.794 | 0.914 | 0.884 |
| | Firmness | 35.553 | 1.487 | 0.952 | 0.884 |
| | Moisture content | 34.895 | 1.118 | 0.927 | 0.879 |
| | L* | 36.594 | 0.638 | 0.929 | 0.643 |
| | a* | 35.585 | 0.867 | 0.920 | 1.353 |
| | b* | 35.797 | 0.488 | 0.923 | 0.032 |

TSS = total soluble solids; $R^2$ = coefficient of determination; RMSE = root mean square error.

### 3.4. Determination of Shelf Life

The high values of $R^2$ indicated that the quality indices have strong temperature dependencies, which can be used as a key indicator in developing shelf life prediction [43]. Firmness and moisture content were selected as the quality indicator for pineapple, signifying that those parameters were suitable to estimate the shelf life of pineapples. In an effort to determine the expressions that could predict the shelf life of pineapples in regard to the changes in firmness and moisture content, the predicted variable time was calculated with respect to the storage temperature. For this reason, the shelf life was assessed as the number of days until the quality changes showed deterioration of the fruit and induced the symptoms of spoilage. It is noteworthy to mention that the robustness of the kinetic models relied on the feasibility to estimate the shelf life of the fruit with the maximum recommended values [44]. The shelf life prediction of pineapples at different temperatures was obtained by combining the first-order kinetic model with the Arrhenius equation as shown in Equation (5). The shelf life of pineapples was determined by a threshold value of the physicochemical properties of the fruit. Nevertheless, there were no specific threshold values for firmness and moisture content due to the variation in threshold values under different storage conditions [39].

$$tSL = \frac{lnCt - lnCo}{kref \exp\left[\frac{-Ea}{RT}\right]} \tag{5}$$

where $t_{SL}$ is the predicted shelf life of pineapples (days), $C_t$ is the limiting value of physicochemical properties, $C_o$ is the initial value of physicochemical properties, $k_{ref}$ is the reaction

rate at reference temperature (day$^{-1}$) of each physicochemical property, $E_a$ is the activation energy (kJ/mol) of each physicochemical property, and $T$ is the absolute temperature (K).

Table 6 shows the shelf life prediction of pineapples using a regression equation according to first-order kinetics at three different storage temperatures. The variation in firmness and moisture content demonstrated that the storage temperatures implied a huge influence on these physicochemical properties of pineapples. The shelf life prediction for MD2 at 5 °C was 31.57 days and 33.58 days based on the firmness and moisture content of the fruit. The shelf life prediction at 10 °C was 39.52 days and 38.76 days in relation to the firmness and moisture content, respectively. Meanwhile, the shelf life prediction observed at 25 °C was 26.52 days and 28.41 days based on firmness and moisture content, respectively. For Josapine, the longest shelf life was 27.77 (firmness) and 29.26 days (moisture content), both recorded at 10 °C. Among all the varieties, Morris had the highest predicted shelf life based on firmness (34.12 days) and moisture content (32.96 days) at 10 °C, respectively. Specifically, a linear model described well the changes in shelf life in terms of firmness and moisture content, achieving an $R^2$ value greater than 0.82 for all pineapple varieties.

**Table 6.** Shelf life prediction of pineapples using regression equation at different storage temperatures in relation to the changes in firmness and moisture content.

| Variety | Physicochemical Properties | Temperature (°C) | Regression Equation | $R^2$ | Predicted Shelf Life (Days) |
|---|---|---|---|---|---|
| MD2 | Firmness | 5 | ln k = −0.053(1/T) + 4.45 | 0.8930 | 31.57 |
| | | 10 | ln k = −0.063(1/T) + 4.95 | 0.9232 | 39.52 |
| | | 25 | ln k = −0.065(1/T) + 3.91 | 0.8942 | 26.52 |
| | Moisture content | 5 | ln k = 2.044(1/T) + 74.04 | 0.9421 | 33.58 |
| | | 10 | ln k = 3.592(1/T) + 72.98 | 0.9884 | 38.76 |
| | | 25 | ln k = 2.824(1/T) + 79.87 | 0.9138 | 28.41 |
| Josapine | Firmness | 5 | ln k = −0.089(1/T) + 3.00 | 0.8970 | 21.52 |
| | | 10 | ln k = −0.096(1/T) + 3.05 | 0.9996 | 27.77 |
| | | 25 | ln k = −0.082(1/T) + 3.13 | 0.9317 | 16.68 |
| | Moisture content | 5 | ln k = 1.239(1/T) + 63.69 | 0.8840 | 25.83 |
| | | 10 | ln k = 1.121(1/T) + 64.76 | 0.8980 | 29.26 |
| | | 25 | ln k = 1.079(1/T) + 66.38 | 0.8812 | 21.19 |
| Morris | Firmness | 5 | ln k = −0.095(1/T) + 4.92 | 0.9233 | 32.53 |
| | | 10 | ln k = −1.026(1/T) + 5.90 | 0.9535 | 34.12 |
| | | 25 | ln k = −1.115(1/T) + 5.24 | 0.9014 | 28.53 |
| | Moisture content | 5 | ln k = 3.636(1/T) + 79.35 | 0.8836 | 29.18 |
| | | 10 | ln k = 3.852(1/T) + 73.36 | 0.9323 | 32.96 |
| | | 25 | ln k = 3.954(1/T) + 69.60 | 0.8968 | 28.27 |

$R^2$ = coefficient of determination.

It can be noted that the samples stored at 10 °C had the longest shelf life, followed by the samples stored at 5 and 25 °C. In this study, the shelf life prediction models for pineapples were established based on the key quality indicators that could provide a theoretical basis for real-time monitoring of quality parameter changes in pineapples. In view of monitoring the quality and safety of fruits, the shelf life prediction was determined according to the different storage temperatures [45]. In this sense, it could be concluded that the prediction model is an alternative approach to determine the shelf life of pineapples as well as to provide consumers with information regarding the storage conditions of fruit supply. It becomes apparent that it is essential to determine the specific factors for shelf life estimation considering that the decision generates different possible outcomes.

## 4. Conclusions

The developed kinetic models based on the changes in physicochemical properties of different pineapple varieties stored at different storage temperatures deliver rapid

information with regard to the shelf life of the fruit, which ultimately can be beneficial for commercial scale. In this study, TSS, pH, moisture content, firmness, and colour evaluation of the MD2, Josapine, and Morris pineapple varieties were investigated during storage. The changes in physicochemical properties of different pineapple varieties were best described by the first-order reaction kinetic model. The storage temperatures highly influenced the variation in quality changes in the fruit over the storage days, with the longest shelf life recorded at 10 °C, compared with 5 and 25 °C. According to the trend of physicochemical properties of pineapple in all tested varieties, the shelf life prediction of the fruit based on the storage temperatures indicated that the Arrhenius model can deliver an efficient tool for monitoring the quality of fruit using a shorter calculation time. This study could provide the feasibility of shelf life prediction for quality monitoring of fresh fruits and vegetables at different storage temperatures during storage. Further research is required on the application of the shelf life prediction model for other varieties and storage conditions of pineapple with subtle chemical and sensorial differences.

**Author Contributions:** Data curation, M.M.A.; investigation, M.M.A.; software, M.M.A.; writing—original draft preparation, M.M.A.; visualisation, M.M.A.; supervision, N.H., S.A.A., and O.L.; validation, N.H.; writing—reviewing and editing, N.H.; formal analysis, S.A.A.; conceptualisation, O.L. All authors have read and agreed to the published version of the manuscript.

**Funding:** This work was funded by the Putra Grant, GP-IPB (Project code: GP-IPB/2020/9687800).

**Data Availability Statement:** Not applicable.

**Acknowledgments:** The authors are thankful for the support and facilities provided by the Department of Biological and Agricultural Engineering, Faculty of Engineering, Universiti Putra Malaysia.

**Conflicts of Interest:** The authors declare no conflict of interest.

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
