# Peer review of "Shelf Life Prediction and Kinetics of Quality Changes in Pineapple (Ananas comosus) Varieties at Different Storage Temperatures"

_horticulturae, doi:10.3390/horticulturae8110992_

Round 1

Reviewer 1 Report (Previous Reviewer 2)

The authors have accepted my suggestions and all comments have been
revised accordingly. I suggest accepting the manuscript. 

Author Response

Thank you for the comments and suggestions.

Reviewer 2 Report (Previous Reviewer 3)

The work is the second version I got for review. Many comments were taken on board and corrected or changed. After analyzing the work, I still have some doubts / questions:

Part of "Materials and Methods"

Methods described correctly and clearly, including sample preparation. The description indicates that before placing the samples at different temperatures for 21 days, all the fruits were first stored for 24 hours at room temperature and equal humidity. This is clearly explained, and I have no doubts here. Why, then, already in the result part, there are such large differences in the physicochemical parameters of the fruit in individual temperature variants on day 0, since these are the same fruit from the preliminary conditions -  STILL NOT COMMENTED ON BY THE AUTHORS. It is difficult to analyze changes between samples, since already on day "0" TSS, pH, firmness and moisture values differ by up to 100% (for example table 2, Morris variety, firmness values on day “zero” are 1.59; 2.92 and 1.27).

Part of Results and Discussion

1)      STILL MY COMMENT CONCERNS THE LACK OF HOMOGENEITY OF THE SAMPLES AT THE BEGINNING OF THE EXPERIMENT ON DAY ZERO. - Referring to the description of the preparation of the samples (lines 93-94), that all fruits were first "stabilized" for 24 hours under the same conditions, and then divided into three temperature variants, my doubts are raised by such different values ​​of parameters for all initial samples, i.e. " day 0 ”, as shown in tables 1 and 2. There are obvious differences, resulting from the natural variability of the fruit, but in many cases they are differences by several dozen percent, e.g. table 2, the moisture content on day 0 for the Morris variety is: 90.35 ; 68.87 and 84.07. Likewise, the TSS content for Morris on "day 0" is 3.60; 9.20 and 7.80. How will the authors explain this? Perhaps the samples taken were not representative…? It is impossible for the dry substance content (TSS) to differ threefold in the initial tests (3.6 and 9.2). This is a clear methodological error in selecting the size of the samples for analysis.

2)      Tables 1 and 2 present the results of the statistical analysis in the form of the significance of differences between the means (marked with different letters). Unfortunately, the legend below the table does not specify which group of results this analysis relates to. Was the significance of differences for a given parameter compared within a given cultivar or only within a given temperature variant in a given cultivar; in a column or in a row? There is an attempt to explain in the text (lines 165-167), but they only concern the firmness parameter. I do not understand why only this one parameter? And the other parameters were analyzed according to a different principle? Should be improved and explained. Why is there still no such explanation in the legend under Tables 1 and 2 ?  

3)      In point 3.1. the Authors explain the method of statistical analysis for the "firmness" parameter (why only for this parameter, and not for all? Incidentally, the description of the satistic analysis is still unclear). Why is there a comment on firmness first, i.e. the data from Table 2, and then there is a description of the results from Table 1, i.e. TSS and pH. I don't understand where this order came from? The results should be reported in the same order as in the following tables.

Author Response

Point 1: Methods described correctly and clearly, including sample preparation. The description indicates that before placing the samples at different temperatures for 21 days, all the fruits were first stored for 24 hours at room temperature and equal humidity. This is clearly explained, and I have no doubts here. Why, then, already in the result part, there are such large differences in the physicochemical parameters of the fruit in individual temperature variants on day 0, since these are the same fruit from the preliminary conditions -  STILL NOT COMMENTED ON BY THE AUTHORS. It is difficult to analyze changes between samples, since already on day "0" TSS, pH, firmness and moisture values differ by up to 100% (for example table 2, Morris variety, firmness values on day “zero” are 1.59; 2.92 and 1.27).

Response 1: The explanation has been added to the manuscript at L180-184.

Point 2: STILL MY COMMENT CONCERNS THE LACK OF HOMOGENEITY OF THE SAMPLES AT THE BEGINNING OF THE EXPERIMENT ON DAY ZERO. - Referring to the description of the preparation of the samples (lines 93-94), that all fruits were first "stabilized" for 24 hours under the same conditions, and then divided into three temperature variants, my doubts are raised by such different values of parameters for all initial samples, i.e. " day 0 ”, as shown in tables 1 and 2. There are obvious differences, resulting from the natural variability of the fruit, but in many cases they are differences by several dozen percent, e.g. table 2, the moisture content on day 0 for the Morris variety is: 90.35 ; 68.87 and 84.07. Likewise, the TSS content for Morris on "day 0" is 3.60; 9.20 and 7.80. How will the authors explain this? Perhaps the samples taken were not representative…? It is impossible for the dry substance content (TSS) to differ threefold in the initial tests (3.6 and 9.2). This is a clear methodological error in selecting the size of the samples for analysis.

Response 2: The explanation has been added to L93-104.

Point 3: Tables 1 and 2 present the results of the statistical analysis in the form of the significance of differences between the means (marked with different letters). Unfortunately, the legend below the table does not specify which group of results this analysis relates to. Was the significance of differences for a given parameter compared within a given cultivar or only within a given temperature variant in a given cultivar; in a column or in a row? There is an attempt to explain in the text (lines 165-167), but they only concern the firmness parameter. I do not understand why only this one parameter? And the other parameters were analyzed according to a different principle? Should be improved and explained. Why is there still no such explanation in the legend under Tables 1 and 2? 

Response 3: The legend below Tables 1 and 2 are added. The value that represents different letters within the same column shows significant differences along the storage days at different storage temperatures using Tukey's test at P<0.05 for all physicochemical properties.

Point 4: In point 3.1. the Authors explain the method of statistical analysis for the "firmness" parameter (why only for this parameter, and not for all? Incidentally, the description of the statistic analysis is still unclear). Why is there a comment on firmness first, i.e. the data from Table 2, and then there is a description of the results from Table 1, i.e. TSS and pH. I don't understand where this order came from? The results should be reported in the same order as in the following tables.

Response 4: The explanation for each parameter is rearranged and revised in the same order in the following tables.

Reviewer 3 Report (New Reviewer)

This research paper developed a shelf life prediction model for three kinds of pineapple stored at three different storage temperatures by analyzing the quality attributes of the fruit. By using first-order kinetics, obtained the coefficient of determination (R2) values of quality changes of pineapples and the best storage temperature with pineapple. Overall, this is an interesting study and provides a better understanding. However, some modifications need to be done before the publication.

1. In the introduction, the limitations of past research should be added and elaborated.

2. When describing the effect of temperature on TTS, pH, moisture content, and hardness, I suggest the author use the line chart to make the data presentation look clearer.

3. Sensory evaluation and the indexes that indicated the shelf life of pineapples should be added to the research.

Author Response

Point 1: In the introduction, the limitations of past research should be added and elaborated.

Response 1: The past research has been added at L74-82.

Point 2: When describing the effect of temperature on TTS, pH, moisture content, and hardness, I suggest the author use the line chart to make the data presentation look clearer.

Response 2: Thank you for the comment. However, we prefer to display the results in table form so that the readers could understand better the decreasing/increasing trends between the storage days and different storage temperatures.

Point 3: Sensory evaluation and the indexes that indicated the shelf life of pineapples should be added to the research.

Response 3: Thank you for this suggestion. It would have been interesting to explore the sensory evaluation aspect. In our study, this would not be possible because it is beyond the scope of this paper which focus on the kinetics of quality changes of pineapple varieties at different storage temperatures. However, new information is added to indicate the threshold value used for shelf life prediction at L403-406.

This manuscript is a resubmission of an earlier submission. The following is a list of the peer review reports and author responses from that submission.

Round 1

Reviewer 1 Report

The authors studied the effect of storage temperature (5, 10, and 25 °C) on the quality of different varieties of pineapples (MD2, Josapine, and Morris).  My comments are as follows:

1) The authors did not mention why they chose those varieties. What are the special characteristics of those varieties to be selected? How do they respond to storage temperature?

2) When you are dealing with low-temperature storage (5 and 10 C), the critical aspect of tropical fruits is they are prone to chilling injury. The prediction of the quality and shelf life of the pineapple solely based on the model was insufficient. According to Table 6, a predicted shelf life day of 25 C of pineapples can go up to 28 days. However, when you observe the external quality of all varieties stored up to 21 days, they are considered unmarketable. The authors also should cut the pineapples to indicate the quality status of the flesh.

3) Some analyses can be done to support your conclusions such as morphological characteristics, chilling injury incidence, internal browning, and biochemical analysis related to pineapple quality.

4) The conclusion is also misleading. Low-temperature storage may lead to chilling injury to the fruits. From the kinetic model and Figure 1, you will end up with the pineapples categorized under unmarketable fruits as you refer to the pineapple quality standard. 

Author Response

Point 1: The authors did not mention why they chose those varieties. What are the special characteristics of those varieties to be selected? How do they respond to storage temperature?

Response 1: The most marketable varieties of pineapples for export in Malaysia were used in this study i.e. MD2, Josapine, and Morris. These varieties of pineapples were selected to address the differences in terms of size, appearance, and skin colour. As for the internal attributes of pineapple, it is highly influenced by storage temperature and type of variety which is essential to determine fresh consumption of the fruit.

Point 2: When you are dealing with low-temperature storage (5 and 10 ⁰C), the critical aspect of tropical fruits is they are prone to chilling injury. The prediction of the quality and shelf life of the pineapple solely based on the model was insufficient. According to Table 6, a predicted shelf life day of 25 ⁰C of pineapples can go up to 28 days. However, when you observe the external quality of all varieties stored up to 21 days, they are considered unmarketable. The authors also should cut the pineapples to indicate the quality status of the flesh.

Response 2: By using developed kinetic models, it is possible to predict in advance the shelf life of pineapples according to a predefined storage temperature and to simulate the trend of the physicochemical properties of the fruit along the storage time. In this study, three week periods are chosen as the storage duration by taking into consideration the postharvest shelf life at optimum temperature of 8-10 ⁰C.

Point 3: Some analyses can be done to support your conclusions such as morphological characteristics, chilling injury incidence, internal browning, and biochemical analysis related to pineapple quality.

Response 3: Thank you for this suggestion. It would have been interesting to explore this aspect. However, in our study, this would not be possible because it is beyond the scope of this paper which focus on the kinetics of quality changes of pineapple varieties at different storage temperatures.

Point 4: The conclusion is also misleading. Low-temperature storage may lead to chilling injury to the fruits. From the kinetic model and Figure 1, you will end up with the pineapples categorized under unmarketable fruits as you refer to the pineapple quality standard.

Response 4: The conclusion has been revised.

Reviewer 2 Report

The objective of the paper titled „Shelf Life Prediction and Kinetics of Quality Changes of Pineapple (Ananas Comosus) Varieties at Different Storage Temperatures“, аccording to the authors, it was to investigate the shelf life and changes in the physicochemical properties of three different varieties of pineapple stored at three different temperatures during a storage period of 21 days. The authors developed a kinetic model based on the changes in the physico-chemical properties of pineapple that provide rapid information regarding the shelf life of the fruit, where it was shown that the Arrhenius model can be an effective tool for monitoring the quality of the fruit with a shorter calculation time.

In my opinion, the objectives of this work are interesting and acceptable, and the results of this research can be used on a commercial scale. The introduction is interesting, it explains very well the problems related to storage and spoilage of pineapples after harvest. In the Materials and Methods section, the methods are explained in detail, which is commendable.

I recommend the authors to continue further studies in this direction, to further develop kinetic models and apply shelf life prediction models for other varieties and storage conditions of pineapples as well as other fruits.

Some corrections are necessary and they are marked in the text using the tool "Comment". I suggest to accept the manuscript after corrections.

Author Response

Point 1: Some corrections are necessary and they are marked in the text using the tool "Comment". I suggest to accept the manuscript after corrections.

Response 1: All the comments have been revised accordingly.

Point 2: L34: Check whether the pineapple is a climacteric or non-climacteric fruit, given the fact that during storage the ripening process continues after harvest.

Response 2: The statement is correct. Pineapple is a non-climacteric fruit.

Point 3: “The highest TSS values among all storage temperatures were achieved at 25 ºC for Josapine (15.60 %) and MD2 (16.60 %), whereas Morris recorded TSS values of 10.10 % at 10 °C.”

Response 3: The highest TSS values among all storage temperatures were achieved at 25 °C for Josapine (16.60 %) and MD2 (15.60 %), whereas Morris recorded TSS values of 10.10 % at 25 °C.

Point 4: “Based on the results, all pineapple varieties had the highest pH recorded at Day 0 (25 °C) which ranged from 2.40 to 4.10 for Josapine, followed by Morris (2.50-3.60) and MD2 (2.40-3.30), respectively.”

Response 4: Based on the results, all pineapple varieties had the highest pH recorded at Day 0 (25 °C) which was 4.10 for Josapine, followed by Morris (3.60) and MD2 (3.30), respectively.

Point 5: “Moreover, it can be observed that storage temperature at 5 °C had the lowest pH compared to the pineapples stored at 10 and 25 °C for all varieties.”

Response 5: Moreover, it can be observed that storage temperature at 5 °C had the lowest pH compared to the pineapples stored at 10 and 25 °C for the varieties MD2 and Josapine, while for the variety Morris the lowest pH values was at 10 °C.

Point 6: “Based on the results, Morris had the highest firmness values at Day 0 (10 °C) which obtained at a range of 0.45 to 2.92 N.”

Response 6: Based on the results, Morris had the highest firmness values at Day 0 (10 °C) which was 2.92 N.

Point 7: “Based on the findings, Josapine had the highest moisture content values at Day 21 (25 °C) which obtained at a range of 85.56 to 95.26 %.”

Response 7: Based on the findings, Josapine had the highest moisture content values at Day 21 (25 °C) which was 95.26 %.

Point 8: “In this study, TSS, pH, moisture content, firmness, and colour evaluation of different pineapple varieties were investigated during storage.” List the names of the varieties tested

Response 8: In this study, TSS, pH, moisture content, firmness, and colour evaluation of MD2, Josapine, and Morris were investigated during storage.

Point 9: “According to the trend of physicochemical properties of pineapple varieties, the shelf life prediction of the fruit based on the storage temperatures indicated that the Arrhenius model can deliver as an efficient tool for monitoring the quality of the fruit using a shorter calculation time.” Add: in all tested varieties

Response 9: According to the trend of physicochemical properties of pineapple in all tested varieties, the shelf life prediction of the fruit based on the storage temperatures indicated that the Arrhenius model can deliver as an efficient tool for monitoring the quality of the fruit using a shorter calculation time.

Reviewer 3 Report

The work is interesting, it deals with current issues related to the quality of fresh fruit and the optimization of their storage conditions. The raw material that is the subject of work is one of the most popular and known all over the world. Due to its sensory qualities, pineapple is valued by consumers and food processors, and canned in syrup is available in almost all stores. Thus, the development of a shelf life prediction system under given conditions is very valuable for both consumers and fruit distributors and processors.

 After analyzing the work, I have a few questions / comments and additions to the text.

Part of "Materials and Methods"

Methods described correctly and clearly, including sample preparation. The description indicates that before placing the samples at different temperatures for 21 days, all the fruits were first stored for 24 hours at room temperature and equal humidity. This is clearly explained, and I have no doubts here. Why, then, already in the result part, there are such large differences in the physicochemical parameters of the fruit in individual temperature variants on day 0, since these are the same fruit from the preliminary conditions.

Part of Results and Discussion

 1) Referring to the description of the preparation of the samples (lines 93-94), that all fruits were first "stabilized" for 24 hours under the same conditions, and then divided into three temperature variants, my doubts are raised by such different values ​​of parameters for all initial samples, i.e. " day 0 ”, as shown in tables 1 and 2. There are obvious differences, resulting from the natural variability of the fruit, but in many cases they are differences by several dozen percent, e.g. table 2, the moisture content on day 0 for the Morris variety is: 90.35 ; 68.87 and 84.07. Likewise, the TSS content for Morris on "day 0" is 3.60; 9.20 and 7.80. How will the authors explain this? Perhaps the samples taken were not representative…? It is impossible for the dry substance content (TSS) to differ threefold in the initial tests (3.6 and 9.2). This is a clear methodological error in selecting the size of the samples for analysis.

2) Tables 1 and 2 present the results of the statistical analysis in the form of the significance of differences between the means (marked with different letters). Unfortunately, the legend below the table does not specify which group of results this analysis relates to. Was the significance of differences for a given parameter compared within a given cultivar or only within a given temperature variant in a given cultivar? For example, in Table 1, the MD2 variant, TSS content at 10 degrees, the result 11.30 (day 0) and 11.40 (day 21) are marked with different letters, i.e. they differ significantly, and the TSS value 11.30 and 13, 70 no different? The significance markings for differ are therefore either incorrect or not clearly described. Should be improved and explained.

3) The description of the results in Table 4 (lines 319-322) is inconsistent with the data in the table. The text states that the "k" values ​​for a temperature of 5 degrees are high compared to a temperature of 10 and 25 degrees. The data in the table does not confirm this. The "k" values ​​for the temperature of 5 degrees are different, sometimes higher and sometimes lower than for the other temperatures. Please explain that.

Author Response

Point 1: Methods described correctly and clearly, including sample preparation. The description indicates that before placing the samples at different temperatures for 21 days, all the fruits were first stored for 24 hours at room temperature and equal humidity. This is clearly explained, and I have no doubts here. Why, then, already in the result part, there are such large differences in the physicochemical parameters of the fruit in individual temperature variants on day 0, since these are the same fruit from the preliminary conditions.

Response 1: Considering the fact that pineapple is a non-climacteric fruit, the quality changes of the fruit varies and are not uniform. Generally, different pineapple varieties have different unique traits and characteristics. For this reason, pineapples are evaluated based on physicochemical attributes of different varieties with acceptable flavour and morphological characteristics.

Point 2: Referring to the description of the preparation of the samples (lines 93-94), that all fruits were first "stabilized" for 24 hours under the same conditions, and then divided into three temperature variants, my doubts are raised by such different values of parameters for all initial samples, i.e. " day 0 ”, as shown in tables 1 and 2. There are obvious differences, resulting from the natural variability of the fruit, but in many cases they are differences by several dozen percent, e.g. table 2, the moisture content on day 0 for the Morris variety is: 90.35 ; 68.87 and 84.07. Likewise, the TSS content for Morris on "day 0" is 3.60; 9.20 and 7.80. How will the authors explain this? Perhaps the samples taken were not representative…? It is impossible for the dry substance content (TSS) to differ threefold in the initial tests (3.6 and 9.2). This is a clear methodological error in selecting the size of the samples for analysis.

Response 2: The pineapple samples were harvested on the same day to avoid the seasonal variances on the physicochemical properties between the varieties. The fruit samples at a maturity level of Index 2 were harvested to evaluate the quality changes of the fruit for a period of three weeks. These pineapple varieties were chosen based on homogenous in size, shape, and grade according to Federal Agricultural Marketing Authority (FAMA). The selected fruit had a medium size and 1st grade quality with an average weight of 1.0-1.2 kg. For instance, these findings were in agreement with Dolhaji et al. (2019) who reported the TSS content ranged from 8.8 to 16.0 % from three pineapple varieties (MD2, Josapine, and Morris).

Point 3: Tables 1 and 2 present the results of the statistical analysis in the form of the significance of differences between the means (marked with different letters). Unfortunately, the legend below the table does not specify which group of results this analysis relates to. Was the significance of differences for a given parameter compared within a given cultivar or only within a given temperature variant in a given cultivar? For example, in Table 1, the MD2 variant, TSS content at 10 degrees, the result 11.30 (day 0) and 11.40 (day 21) are marked with different letters, i.e. they differ significantly, and the TSS value 11.30 and 13, 70 no different? The significance markings for differ are therefore either incorrect or not clearly described. Should be improved and explained.

Response 3: The mean comparison of each physicochemical property along the storage days at different pineapple varieties was analysed using Tukey’s test. The value that represents different letters within the same column shows significant differences along the storage days at different storage temperatures. Explanations are added to the text.

Point 4: The description of the results in Table 4 (lines 319-322) is inconsistent with the data in the table. The text states that the "k" values for a temperature of 5 degrees are high compared to a temperature of 10 and 25 degrees. The data in the table does not confirm this. The "k" values for the temperature of 5 degrees are different, sometimes higher and sometimes lower than for the other temperatures. Please explain that.

Response 4: The sentence has been revised. Explanation has been added.